# Crystal structure of Mokola virus glycoprotein in its post-fusion conformation

**Laura Belot**[1], **Malika Ouldali**[1], **Stéphane Roche**[1], **Pierre Legrand**[2], **Yves Gaudin**[1]\*, **Aurélie A. Albertini**[1]\*

1 Institute for Integrative Biology of the Cell (I2BC), CEA, CNRS, Univ. Paris-Sud, Université Paris-Saclay, France, 2 Synchrotron SOLEIL, France

\* yves.gaudin@i2bc.paris-saclay.fr (YG); aurelie.albertini@i2bc.paris-saclay.fr (AAA)

## Abstract

Mokola virus (MOKV) belongs to the lyssavirus genus. As other genus members—including rabies virus (RABV)—it causes deadly encephalitis in mammals. MOKV entry into host cells is mediated by its transmembrane glycoprotein G. First, G binds cellular receptors, triggering virion endocytosis. Then, in the acidic endosomal environment, G undergoes a conformational change from its pre- toward its post-fusion state that catalyzes the merger of the viral and endosomal membranes. Here, we have determined the crystal structure of a soluble MOKV G ectodomain in which the hydrophobic fusion loops have been replaced by more hydrophilic sequences. The crystal structure corresponds to a monomer that is similar to the protomer of the trimeric post-fusion state of vesicular stomatitis virus (VSV) G. However, by electron microscopy, we show that, at low pH, at the surface of pseudotyped VSV, MOKV spikes adopt the trimeric post-fusion conformation and have a tendency to reorganize into regular arrays. Sequence alignment between MOKV G and RABV G allows a precise location of RABV G antigenic sites. Repositioning MOKV G domains on VSV G pre-fusion structure reveals that antigenic sites are located in the most exposed part of the molecule in its pre-fusion conformation and are therefore very accessible to antibodies. Furthermore, the structure allows the identification of pH-sensitive molecular switches. Specifically, the long helix, which constitutes the core of the post-fusion trimer for class III fusion glycoproteins, contains many acidic residues located at the trimeric interface. Several of them, aligned along the helix, point toward the trimer axis. They have to be protonated for the post-fusion trimer to be stable. At high pH, when they are negatively charged, they destabilize the interface, which explains the conformational change reversibility. Finally, the present structure will be of great help to perform rational mutagenesis on lyssavirus glycoproteins.

## Author summary

Mokola virus (MOKV), as other lyssaviruses—including rabies virus (RABV)—causes deadly encephalitis in mammals. Its envelope glycoprotein G is involved in two successive steps of virus entry: receptor recognition, to allow virion endocytosis, and fusion of viral and endosomal membranes, to release the viral nucleocapsid into the cytoplasm for the

**Data Availability Statement:** The data underlying the results will be available at the PDB (access code: 6TMR).

**Funding:** This work was supported by the CNRS and a grant from the Agence Nationale de la

Recherche (ANR CE11 MOBARHE) to Yves Gaudin. The IMAGIF integrated platform was financially supported by the French Infrastructure for Integrated Structural Biology Initiative (FRISBI, ANR 10 INSB 05 02). The funders had no role in study design, data collection and analysis, decision to publish, or preparation of the manuscript.

**Competing interests:** The authors have declared that no competing interests exist.

subsequent steps of infection. Fusion is triggered by a low pH-driven conformational change of G from its pre-fusion toward its post-fusion conformation. G is also the target of neutralizing antibodies. In our investigation, we determined the crystal structure of MOKV G in its post-fusion conformation, which allows the precise location of the antigenic sites of lyssaviruses glycoproteins that are exposed at the top of the molecule in a model of MOKV G pre-fusion state. We also identified several acidic residues and two histidines that play the role of pH-sensitive molecular switches during the structural transition. This first lyssavirus glycoprotein structure may be useful for teams working on entry and antigenicity of the highly pathogenic viruses of this genus.

## Introduction

Lyssavirus is a genus of the Rhabdoviridae family, which consists of 16 species. All lyssaviruses cause acute progressive encephalitis (rabies disease) in mammals and are transmitted between susceptible individuals directly by bites, scratches, or contamination of mucous membranes with infected saliva. They have been classified into three phylogroups: the prototype of phylogroup I is rabies virus (RABV), the prototype of phylogroup II is Mokola virus (MOKV), and the prototype of phylogroup III is West Caucasian bat lyssavirus (WCBV) [1].

Lyssaviruses are enveloped viruses whose genome encodes a single transmembrane glycoprotein (G) constituting the spikes that protrude at the viral surface. G plays a key role in the initial stages of infection [2,3]. First, it binds cellular receptors [4–7]. Subsequently, the viral particle enters the cell via the endocytic pathway [8,9]. Then, the acidic environment of the endosome induces a huge conformational change of G from a native (or pre-fusion) state toward a final (or post-fusion) state that catalyzes fusion of the viral envelope with the endosomal membrane [10]. This results in the release of the nucleocapsid in the cytoplasm for the subsequent steps of infection.

The glycoprotein G is a type I glycoprotein. After cleavage of the N-terminal signal peptide, the mature glycoprotein is about 500 amino acid residues long (505 for RABV, 503 for MOKV). It is anchored in the membrane by a single α-helical transmembrane segment. The bulk of the mass of G is located outside the viral membrane and constitutes the N-terminal ectodomain that is N-glycosylated (Fig 1A). The number of glycosylation sites may vary from one lyssavirus to another.

At the surface of the virus, it has been shown that RABV G is organized in trimers [11]. At pH 7.3, the spikes are in their pre-fusion conformation and form a well-defined layer that is 8 nm thick at the virion surface [11]. At low pH (below pH 6.4), a jagged layer of elongated spikes (~ 11.5 nm) in their post-fusion state is observed [10]. At low pH, RABV virions have also a strong tendency to aggregate [10], a feature which is presumably due to transient exposition of G fusion loops that are hydrophobic motifs interacting with the target membrane as a first step of the fusion process [12]. Remarkably, G structural transition is reversible and there is an equilibrium between different states of G, which is shifted toward the post-fusion conformation at low pH [10,13].

G ectodomain is the target of neutralizing antibodies [14–17]. Several hundred monoclonal antibodies (MAbs) have been used to characterize the antigenic structure of RABV G [14,16,18–20], which has two major antigenic sites (antigenic site II and antigenic site III) [19,20]. Antigenic site III (extending from residues 330 to 338) is associated with virulence [20,21]. In particular, RABV antigenic mutants having amino acid substitutions at position 333 have an avirulent phenotype [20–22], and the presence of a lysine and an arginine

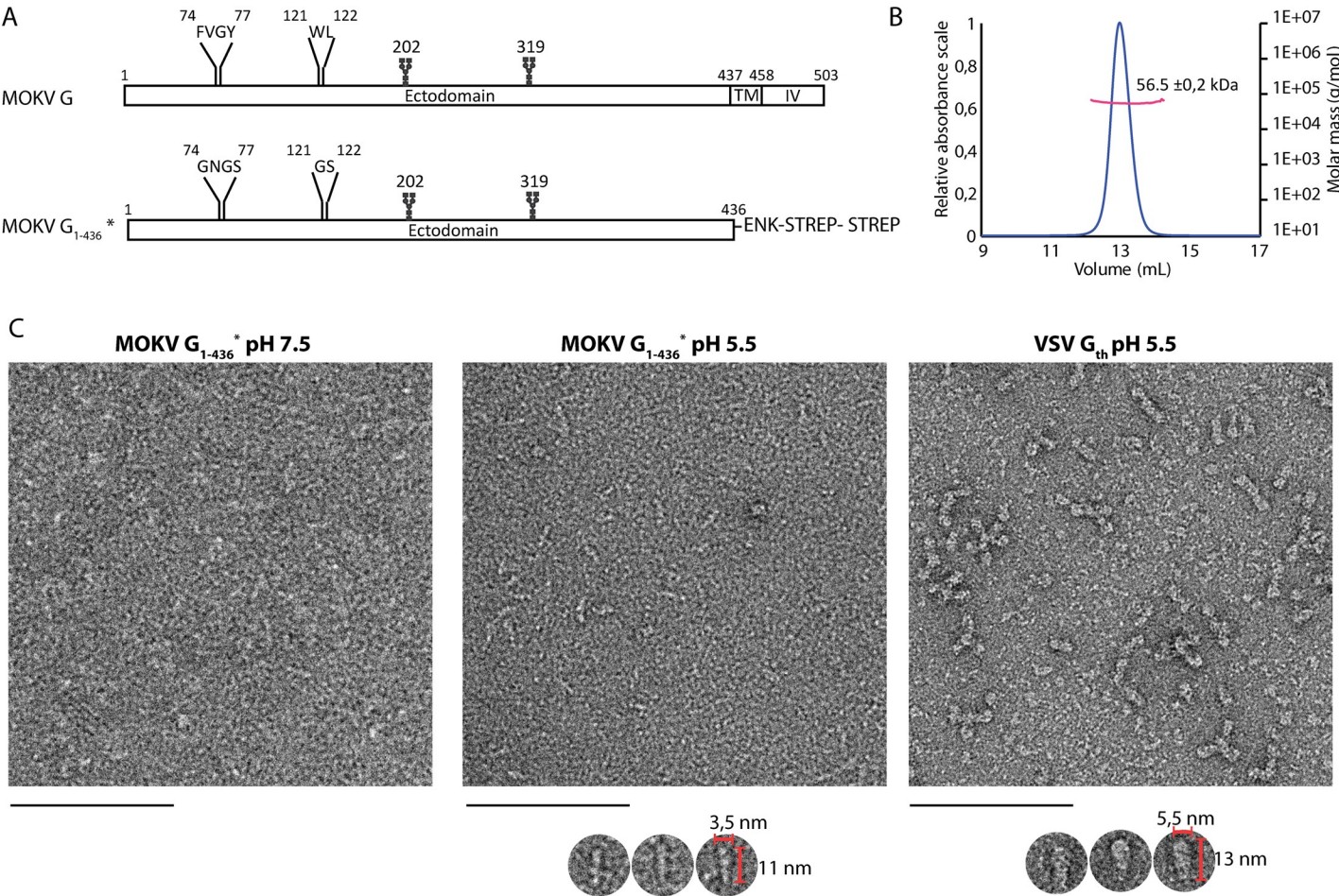

**Fig 1. Construction, purification, and characterization of MOKV $G_{1-436}*$.** A) Bar diagram showing the organization of MOKV G and $G_{1-436}*$. The N-glycosylation sites are indicated as well as the two fusion loops in G and the hydrophilic sequences that replace them in $G_{1-436}*$. TM: transmembrane domain; IV: intraviral domain; ENK: enterokinase cleavage site; STREP: StrepTag. B) Molecular mass of MOKV $G_{1-436}*$ determined by SEC-MALS. C) Electron microscopy on negatively stained MOKV $G_{1-436}*$ at pH 7.5 and 5.5 and VSV $G_{th}$ at pH 5.5. At pH 7.5, no defined molecular shape of MOKV $G_{1-436}*$ can be identified. At pH 5.5, elongated structures can be distinguished as shown in the gallery below the micrograph. These structures are thinner than those observed with the post-fusion $G_{th}$ trimers. Scale bar: 100 nm.

respectively in positions 330 and 333 of RABV G is required for virus penetration into the motor and sensory neurons of adult mice [22]. Finally, in addition to these major antigenic sites, one minor antigenic site (minor antigenic site a) [14] and a few isolated epitopes [18,23] have been described.

Sequence similarity and identity indicate that lyssaviruses glycoproteins adopt the same fold as other rhabdovirus glycoproteins and therefore belong to the third class (class III) of viral fusion glycoproteins. However, in the rhabdovirus family, only crystal structures of glycoproteins belonging to members of the vesiculovirus genus have been determined. They include the trimeric pre-fusion state of vesicular stomatitis virus (VSV) G [24], the trimeric post-fusion state of VSV and Chandipura virus (CHAV) G [25,26], and monomeric states of CHAV G corresponding to early and late intermediates during the structural transition pathway [27].

Here, we present the crystal structure of a soluble MOKV G ectodomain, expressed in *Drosophila* Schneider 2 (S2) cells. A comparison with vesiculovirus glycoprotein structures reveals that the crystal contains a monomeric form of G having the organization of a protomer of the post-fusion trimer. Using negative staining electron microscopy (EM), we detected post-fusion

trimers of soluble MOKV G ectodomain interacting with membranes as well as full-length MOKV G at the surface of VSV pseudotypes. EM also revealed that as VSV G, MOKV G in its post-fusion conformation forms regular networks. Finally, the MOKV G crystal structure allows the precise localization of the antigenic sites of lyssavirus glycoproteins and provides the molecular basis of the reversibility of the conformational change.

## Results

### Expression, purification, and characterization of MOKV G ectodomain

We synthesized two genes encoding MOKV G ectodomain (residues 1–436) C-terminally fused to an enterokinase-cleavable double StrepTag. The first gene corresponds to the wild type ectodomain sequence ($G_{1-436}$WT). The second one corresponds to a construct in which the hydrophobic fusion loops $^{74}$FVGY$^{77}$ and $^{121}$WL$^{122}$ have been replaced by more hydrophilic sequences $^{74}$GNGS$^{77}$ and $^{121}$GS$^{122}$ ($G_{1-436}$*) (Fig 1A).

Both constructs were expressed using stably transfected S2 cells. The production yield of $G_{1-436}$* was much higher than $G_{1-436}$WT. Furthermore, the purity of $G_{1-436}$* after affinity chromatography was much better than that of $G_{1-436}$WT (S1 Fig). As a consequence, only $G_{1-436}$*

**Table 1. Data collections and refinement statistics.**

| Data collection | |
|---|---|
| Wavelength | 1.648 |
| Resolution range | 45.22–2.893 (2.997–2.893) |
| Space group | C2221 |
| Unit cell | a = 76.279, b = 81.951, c = 231.337 |
| Total reflections | 216607 (31149) |
| Unique reflections | 15237 (885) |
| Completeness (%) | 91.60 (54.07) |
| Mean I/sigma(I) | 10.16 (0.95) |
| CC1/2 | 0.998 (0.798) |
| **Refinement statistics** | |
| Reflections used in refinement | 15234 (884) |
| Reflections used for R-free | 783 (47) |
| R-work | 0.2360 (0.3475) |
| R-free | 0.2852 (0.4107) |
| Number of non-hydrogen atoms | 3144 |
| macromolecules | 3124 |
| ligands | 20 |
| Protein residues | 394 |
| RMS (bonds) | 0.011 |
| RMS (angles) | 1.40 |
| Ramachandran favored (%) | 93.83 |
| Ramachandran allowed (%) | 5.14 |
| Ramachandran outliers (%) | 1.03 |
| Rotamer outliers (%) | 0.56 |
| Clashscore | 16.02 |
| Average B-factor | 76.86 |
| macromolecules | 76.55 |
| ligands | 125.24 |
| PDB ID | 6TMR |

was further characterized. The mass of the protein was determined by SEC-MALS (Fig 1B). At pH 8, the molecular mass of 56.5 +/- 0.2 kDa is constant across the elution peak and is consistent with the mass of a monomer of $G_{1-436}$* (52682 Da) having two glycosylation sites. At pH 6, the protein remains stuck on the column impeding its mass determination.

The structure of $G_{1-436}$* was also analyzed by negative staining EM at pH 7.5 and 5.5 (Fig 1C). At pH 7.5, no defined molecular shape could be identified due to the small size or flexibility of monomeric $G_{1-436}$*. At pH 5.5, elongated structures were observed. However, they appear thinner than VSV $G_{th}$ (aa residues 1–422, VSV G fragment that was previously crystallized [24,26]) *bona fide* post-fusion trimers, suggesting that they correspond to extended monomers.

## Crystal structure of MOKV $G_{1-436}$*

We screened several crystallization conditions for MOKV $G_{1-436}$*, obtained crystals at pH 7.5 and determined the structure of the protein at 2.9 Å resolution by single-wavelength anomalous dispersion (SAD) (Table 1). The asymmetric unit contained a single protein. The chain was traced from residues 1 to 109 and 118 to 401 (Fig 2A). Although the crystal structure is not trimeric, the overall structure is very similar to the protomer of the trimeric post-fusion state of CHAV and VSV G (Fig 2B). In particular, the lateral helix is already formed (Fig 2C) and the conserved residues D143, Y145, and H397 that lock the hairpin-shaped post-fusion conformation are at the same place as in VSV G post-fusion protomer (Fig 2D). This is not the case in the late intermediate (LI) conformation of CHAV G (Fig 2C and 2D).

Structure-based sequence alignment (S2 Fig) allowed a precise delimitation of the fusion domain (FD), the pleckstrin homology domain (PHD), and the trimerization domain (TrD) (Fig 2B, Fig 3 and Table 2). As in vesiculovirus glycoproteins, these domains are flanked by segments (R1 to R5) that refold during the low pH-induced conformational change.

The FD is extremely well conserved among rhabdoviruses (Fig 3A) with two insertions in MOKV G sequence (residues 113–116 and 162–166) (S2 Fig) that are also present in other lyssaviruses G (S3 Fig). In this domain, the segment 110–117, containing sequence insertion 113–116, is not visible, which indicates that it is more flexible in lyssaviruses than in vesiculoviruses. Indeed, lyssavirus glycoproteins do not have a stabilizing disulfide bridge equivalent to that found at the tip of the vesiculovirus FD (between cysteine residues 68 and 114 for VSV G) (Fig 3A).

The structural organization of the TrD is also well conserved except for the most lateral part, which also contains an extra disulfide bridge in lyssavirus G (between cysteine residues 344 and 351) (Fig 3B). However, the long helix, which constitutes the center of the post-fusion trimer for all class III fusion glycoproteins, is bent in MOKV G, which is not the case for vesiculoviruses G (Fig 3B). This indicates that the trimerization motif of MOKV G is a coiled-coil, whereas it is a trimeric bundle of straight helices for vesiculovirus glycoproteins. Finally, the PHD is much less conserved—a feature that was previously observed among the vesiculovirus genus [25] (Fig 3C).

## Electron microscopy reveals post-fusion trimers of MOKV G

As we had not been able to observe post-fusion trimers of $G_{1-436}$*, we made the hypothesis that the formation of the post-fusion trimer would be favored after membrane insertion of the fusion loops into a target membrane. Such insertion is not possible with $G_{1-436}$*. Therefore, we used purified $G_{1-436}$WT that we incubated with liposomes at pH 5.5 for two hours at 37˚C. Although, as mentioned above, $G_{1-436}$WT samples are less pure than $G_{1-436}$*, we observed liposomes nicely decorated by G ectodomains, which clearly adopt the trimeric post-fusion

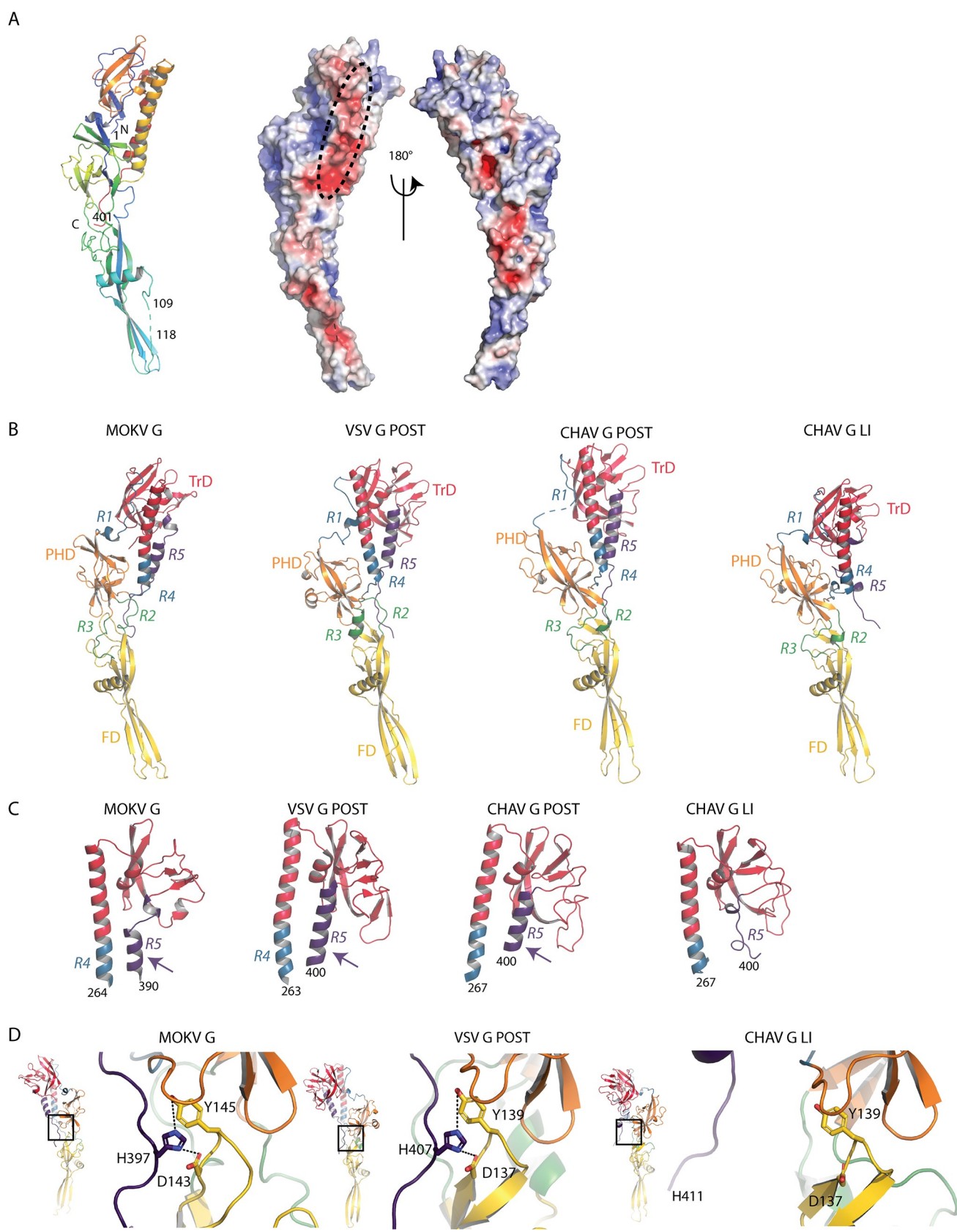

**Fig 2. Structure of MOKV G$_{1-436}$* compared to the post-fusion conformation of VSV and CHAV G$_{th}$ and to the late intermediate (LI) conformation of CHAV G$_{th}$.** A) Left: Ribbon representation of the structure of MOKV G$_{1-436}$*. Rainbow coloring from blue to red indicates the N- to C-terminal position of the residues in the model. Right: electrostatic surface potentials distribution on the solvent-accessible surface of MOKV G1-436* generated using the APBS plugin in Pymol software [55]. The acidic side of the TrD helix is surrounded by a black ellipse. B) Ribbon representation of the structures of MOKV G$_{1-436}$*, of a protomer of the trimeric post-fusion conformations of VSV and CHAV G$_{th}$, and of the monomeric CHAV G$_{th}$ LI colored by domains following the same color code as for MOKV G$_{1-436}$*. In A) and B), the glycoproteins are aligned on their fusion domain. C) Structure of TrD and segments R4 and R5 in MOKV G$_{1-436}$* in a protomer of the trimeric post-fusion conformation of VSV and CHAV G$_{th}$ and in monomeric CHAV G$_{th}$ LI. The arrows indicate the lateral helix that is present in the post-fusion conformation but not in LI conformation. D) Close-up view on the triad D143, Y145, and H397 of MOKV G$_{1-436}$* that locks the hairpin-shaped post-fusion conformation and on their counterpart in the post-fusion protomer of VSV G$_{th}$ and in CHAV G$_{th}$ LI.

conformation inserted in the membrane via their hydrophobic fusion loops (Fig 4A). This demonstrated that the enterokinase cleavage site and the double StrepTag, which are also present in G$_{1-436}$WT, do not impede G trimerization. Furthermore, as with VSV G ectodomain [28], we observed the formation of long proteolipidic tubular structures having a diameter of ~ 40 nm (Fig 4B). Cryo electron-microscopy revealed that, at the surface of these structures, G trimers are organized in a regular helical network (Fig 4C) having a pitch of ~ 6.25 nm.

We also investigated the structure of full-length wild type MOKV G (MOKV G WT) and mutant MOKV G* (in which the fusion loops were replaced by hydrophilic sequences as in G$_{1-436}$*) at the surface of pseudotyped VSVΔG-eGFP (Fig 5).

At pH 7.5, MOKV G WT can be seen in side view on the periphery of the virions (Fig 5A). The glycoproteins form a well-defined 8-nm-thick layer similar to the one previously observed with RABV G [9–11]. Stain accumulation near the membrane below the head of the protein was consistent with a pre-fusion state similar to the one of vesiculoviruses.

At pH 5.5, the virions are massively aggregated as already described for RABV [10]. At the surface of some isolated virions, the spikes are individualized, allowing the visualization of spikes that stand perpendicular to the membrane (Fig 5B). Measurements of those spikes revealed that they have a length of 11.97 +/- 0.57 nm and a width of 5.31 +/- 0.52 nm ($n$ = 50). Their length is similar to that of VSV G post-fusion trimer (13 nm). Their width is bigger than that of MOKV G crystalline monomer (3.5 nm). Therefore, these spikes correspond to the trimeric post-fusion form of MOKV G. In some circumstances, a regular network of G can also be observed (Fig 5B). At the surface of other virions, or at the periphery of the aggregates, MOKV glycoproteins appear to form a fuzzy heterogeneous layer (Fig 5B) in which some spikes have the typical post-fusion shape, while some others are thinner and oblique to the membrane, most probably corresponding to an elongated monomer.

In the case of MOKV G*, the layer of G around the virion is less defined at pH 7.5, suggesting that the pre-fusion state is partially destabilized (Fig 5C). However, at low pH (Fig 5D), MOKV G* behaves the same way as MOKV G WT, and post-fusion trimers can be observed at the surface of the viral particles. This indicates that the hydrophilic sequences that replace the fusion loops do not impede the formation of the post-fusion trimer.

## Antigenic sites

Sequence alignment between MOKV G and RABV G ectodomain (65% amino acid identity) allows the identification of the location of RABV antigenic sites (S4 Fig).

RABV antigenic mutants resistant to neutralization by site II-specific MAbs have mutations in positions 36, 40, 42, 198, and 200 of G, which are indeed clustered at the surface of the PHD (Fig 6A). Furthermore, a mutation in position 147, located in the FD, confers partial or total resistance to most MAbs recognizing site II [19]. Other mutations allowing resistance to neutralization were located in position 34 (in the segment R1) and 184 (in the segment R3) (19) (Fig 6A). Repositioning MOKV G domains on VSV G pre-fusion structure (Fig 6B) shows that

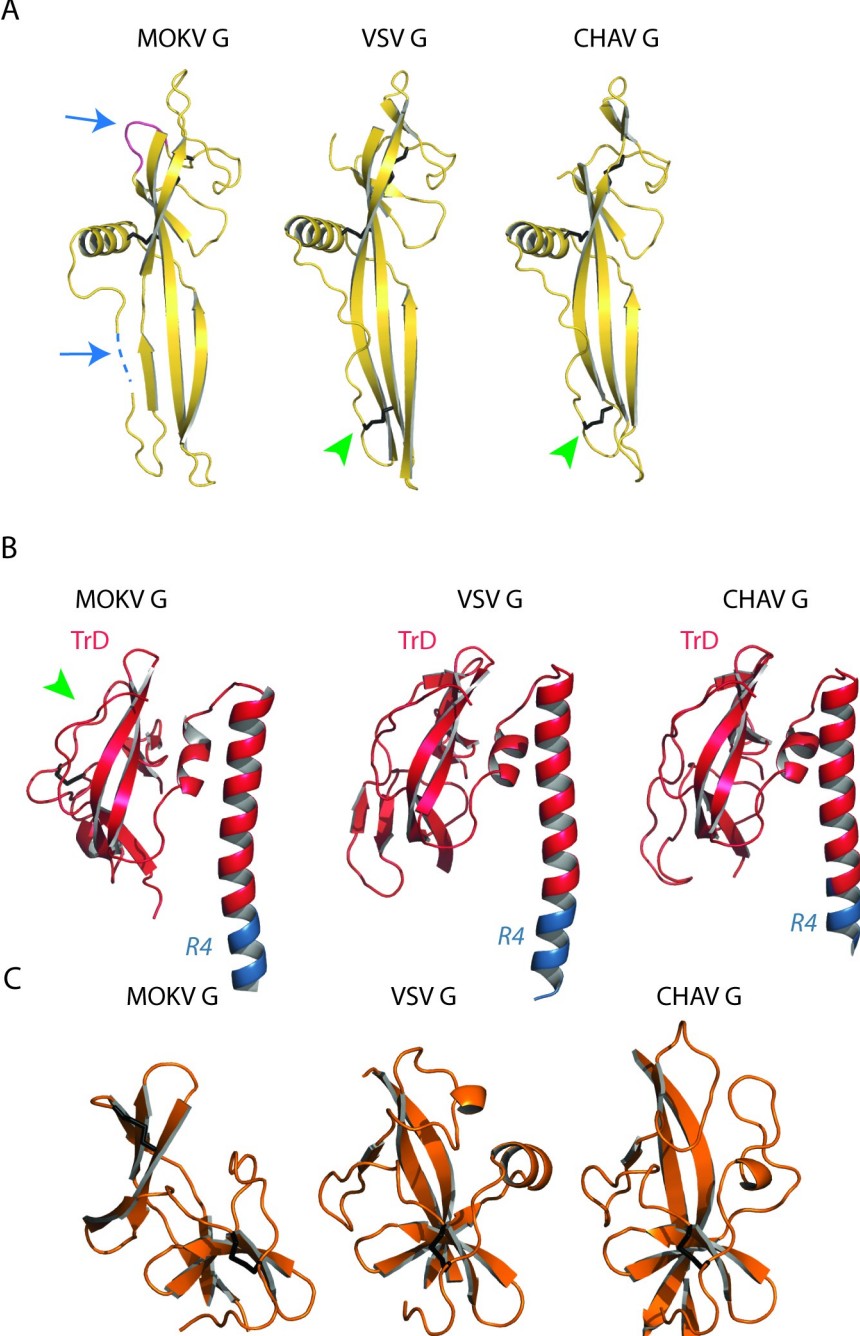

**Fig 3. Comparison of the structure of MOKV G domains with those of VSV and CHAV G.** A) Comparison of FDs of MOKV, VSV, and CHAV G. Blue arrows indicate additional segments present in all lyssaviruses G. The dotted line represents the missing segment in MOKV G structure. Green arrowheads indicate the disulfide bridge present in vesiculoviruses G but not in lyssaviruses G. B) Comparison of TrDs and R4 segment of MOKV, VSV, and CHAV G. Note that MOKV G helix is bent, whereas it is rather straight for CHAV G and VSV G. The green arrowhead indicates the extra disulfide bridge of lyssaviruses G. C) Comparison of PHDs of MOKV, VSV, and CHAV G.

antigenic site II is located at the top of the molecule in its pre-fusion conformation at the viral surface, which certainly explains why it is the major antigenic site of RABV G (14) (Fig 6B and 6C). The fact that site II includes residues belonging to distinct domains and segments R1 and

**Table 2. Domains and segments nomenclature used in the text.**

| Domain | Residues | color |
|---|---|---|
| TrD | 1–17 and 273–373 | Red |
| PHD | 35–46 and 191–259 | Orange |
| FD | 55–180 | Yellow |
| R1 | 18–34 | Blue |
| R2 | 47–54 | Green |
| R3 | 181–190 | Green |
| R4 | 260–272 | Blue |
| R5 | 374–401 | Magenta |

R3 that refold during the structural transition explains why MAbs directed against this site are unable to recognize G in its post-fusion conformation.

RABV antigenic mutants resistant to neutralization by site III-specific MAbs have amino acid substitutions clustered in position 330, 333, 336 and 338 of G [20,21] that are located at the surface of the TrD (Fig 6A) near the top of the molecule in its pre-fusion conformation (Fig 6B and 6C). Several RABV mutants in position 333 have an avirulent phenotype [20,21], and the presence of lysine and arginine respectively in position 330 and 333 of RABV G is required for virus penetration into the neurons of adult mice [22]. Interestingly, in the model of the MOKV G pre-fusion state based on VSV G, those residues (K330 and D333 for MOKV G) are near the interaction site with CR domains from members of the low-density lipoprotein receptor (LDL-R) family [29] which are VSV receptors [30] (Fig 6D). This is consistent with the idea that RABV residues K330 and R333 are involved in RABV receptor(s) recognition.

A minor site (minor site a) had also been identified [14]. Mutants escaping neutralization by MAbs recognizing this minor site were shown to have amino acid substitutions at position 342 and 343 (Fig 6A). Despite their proximity, there is no overlap between site III and minor site a [14], which might be explained by the kink in the polypeptide chain induced by the proline in position 340.

Several other MAbs recognize linear epitopes contained in the region 223–285 [18,23]. Among those MAbs, the one best characterized is certainly MAb 17D2 [18], which recognizes both denatured G and its pre-fusion state [31] and has been shown to recognize region 255–280 [18]. Its epitope largely overlaps with the segment R4 of the post-fusion long helix (S2 Fig) that has no canonical secondary structure in the pre-fusion state (Fig 6C).

Finally, a few MAbs bind only the post-fusion state of RABV G at the viral surface [17,32]. Their epitopes are located in the N-terminal part of the TrD, and neutralization escape mutants have amino acid substitutions in position 10, 13, and 15 [17,32], which are located at the top of the post-fusion conformation (low pH site on Fig 6A). The change of orientation of TrD after the conformational change (Fig 6B) explains why those MAbs are unable to recognize RABV G in its pre-fusion conformation at the viral surface while still being able to bind detergent-solubilized G regardless of its conformational state [31,33].

## pH sensitive molecular switches

The long helix, which constitutes the center of the post-fusion trimer, contains several acidic residues (D263, D266, E267, E269, E274, D275, E281, E282, D285, E288) (Fig 7A). Such residues have been shown to play the role of pH-sensitive molecular switches in VSV G [34].

Inside a protomer, D275 faces H384 (located in the lateral helix of R5), making an interaction that stabilizes the post-fusion state (Fig 7B). E269 faces D211 (located in the PHD), and

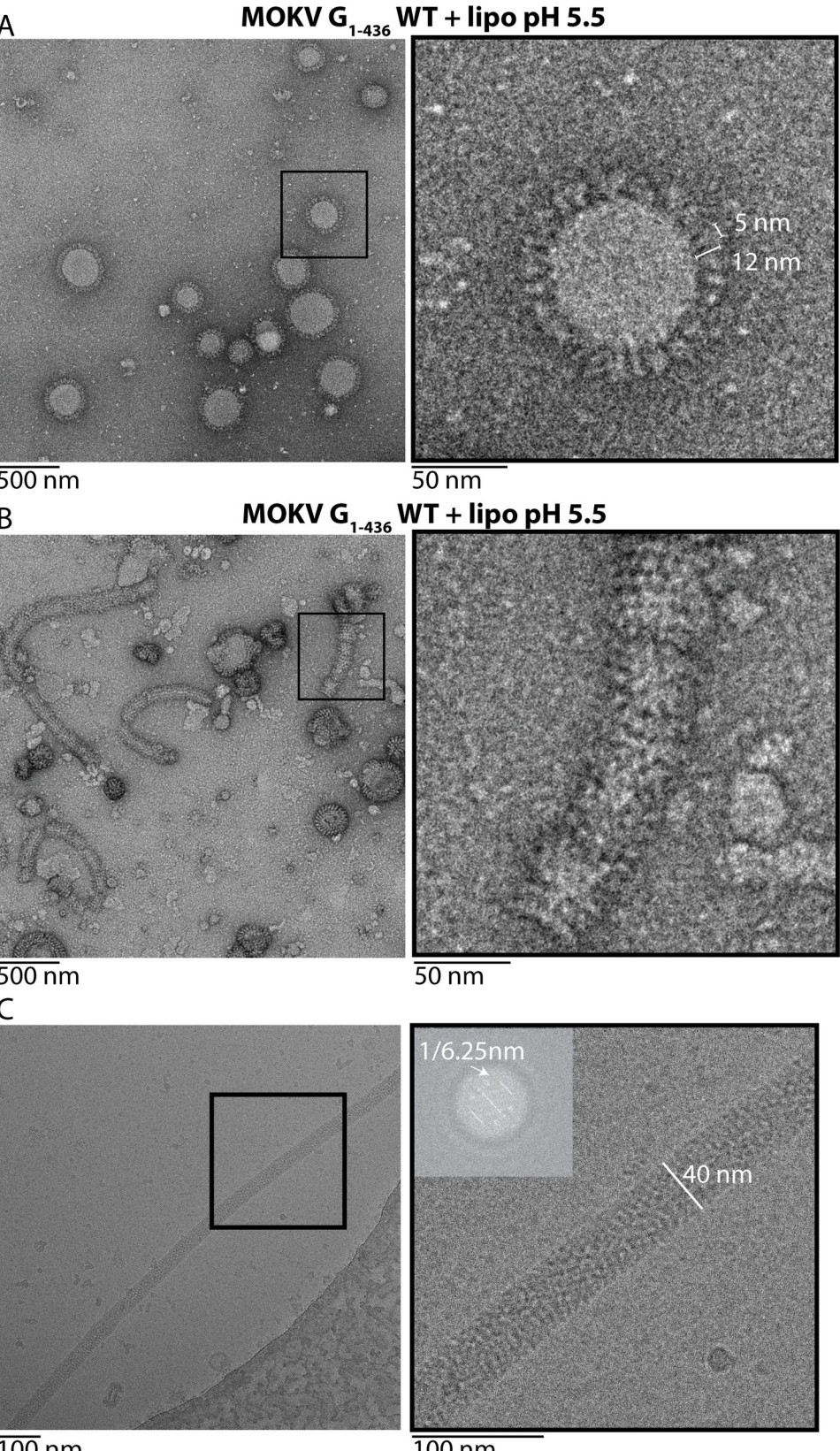

**Fig 4. Electron microscopy of MOKV G$_{1-436}$WT incubated with liposomes at pH 5.5.** A) MOKV G$_{1-436}$WT is inserted in liposomes in a typical trimeric post-fusion conformation. The sample was stained with sodium phosphotungstate (NaPT). B) MOKV G$_{1-436}$WT forms a network favoring the formation of tubular structures. Although these proteolipidic tubes are also seen after NaPT staining, they are more easily observed with uranyl acetate staining that has been used for the micrograph presented in this panel. C) Cryo-EM micrograph of tubular structures formed by MOKV G$_{1-436}$WT when incubated with liposomes at pH 5.5 and corresponding power spectra (diffraction pattern).

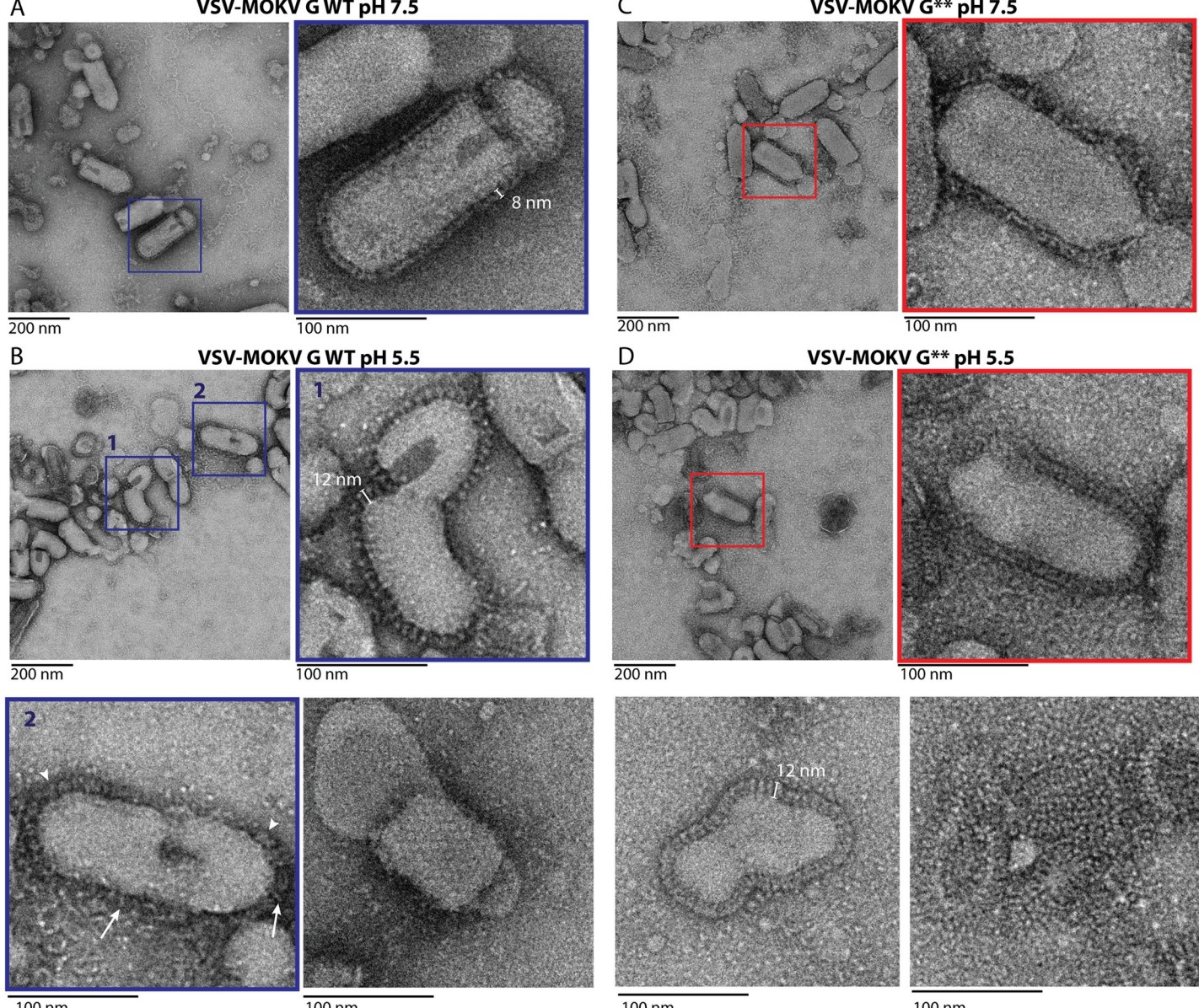

**Fig 5. Electron microscopy of VSVΔG-eGFP pseudotyped either by MOKV G WT or MOKV G***. A) At pH 7.5, VSV-MOKV G WT exhibits an 8-nm-dense layer of glycoproteins at its surface. B) At pH 5.5, some VSV-MOKV G WT virions exhibit 12-nm-long individualized spikes corresponding to the post-fusion trimer that stands perpendicular to the membrane (inset 1). At the surface of some other virions, MOKV G WT appears to form a fuzzy heterogeneous layer in which some spikes have the typical post-fusion shape (arrowheads), while some others are thinner and oblique to the membrane (arrows), most probably corresponding to elongated monomers (inset 2). A regular network of G can also be observed (bottom right micrograph). C) At pH 7.5, at the surface of VSV-MOKV G*, the layer of MOKV G* around the virion is less defined than at the surface of VSV-MOKV G WT. D) At pH 5.5, post-fusion trimers are observed at the surface of VSV-MOKV G* (inset and bottom left micrographs). As with VSV-MOKV G WT, a regular network of MOKV G* can be observed at the surface of some disrupted virions (bottom right micrograph).

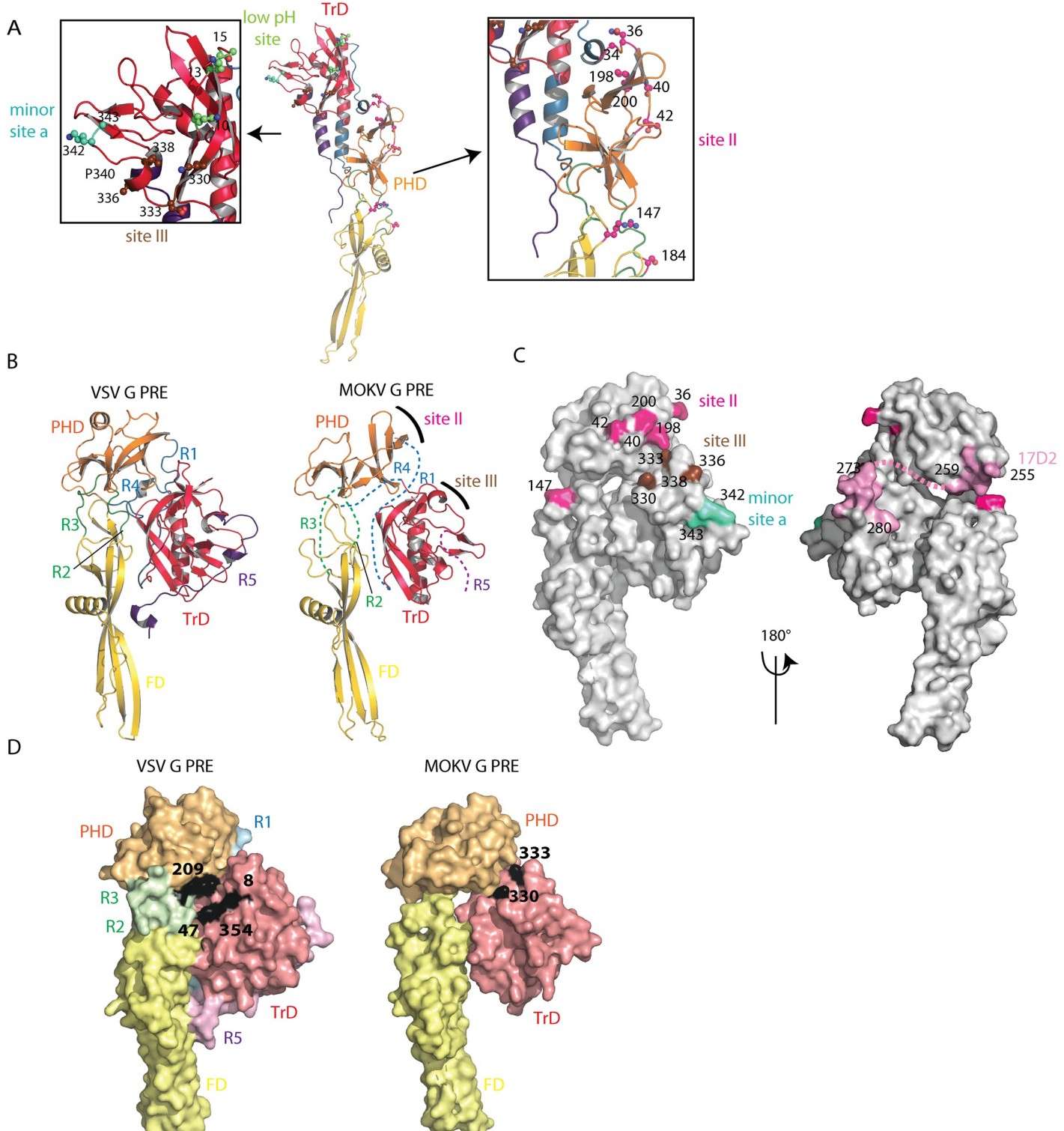

**Fig 6. Antigenic sites or RABV positioned on MOKV G structure.** A) Location of major antigenic sites II and III, minor antigenic site a, and low pH antigenic site. Residues that are found substituted in mutants escaping neutralization by MAbs are indicated by their number and represented in ball and sticks form. They are colored in magenta for site II, in brown for site III, in cyan for minor site a, and in green for low pH antigenic site. Proline 340, which separates site III and minor site a, is also indicated. B) Model of MOKV G pre-fusion generated by superimposing MOKV G PHD, TrD, and FD domains on the corresponding domains of VSV G. The location of major antigenic sites II and III are indicated on the model. C) Surface representation of the modeled MOKV G pre-fusion state. Residues that are

found substituted in mutants escaping neutralization by MAbs are indicated by their number and colored with the same color code as in A. The epitope of Mab 17D2 is colored in pink. Segment R4, which is part of the epitope and of which the pre-fusion conformation is unknown, is indicated by a pink dotted line. D) Surface representation of VSV G and modeled MOKV G pre-fusion states. The footprint of the LDL-R CR2 domain on VSV G pre-fusion conformation is in black (with key residues for binding indicated by their number). The location of residues 330 and 333, required for RABV penetration into neurons of adult mice, are also indicated in black at the surface of MOKV G. Both molecules have the same orientation.

the deprotonation of their side chain at high pH induces a repulsive force that may contribute to the structural transition back to the pre-fusion state (Fig 7C). Finally, residue H261 that is positively charged at low pH is sandwiched between the acidic residues D266 and E45 (Fig 7C). It is worth noting that residues D275, H384, E269, D211, D275, D266, H261, D266 and E45 are all conserved among the lyssavirus genus (with the exception of Irkut virus for which the residue in position 269 is an alanine) (S3 Fig).

The central helix is amphiphilic, with its acidic residues mostly located on the solvent-exposed face—as already indicated by the negative electrostatic potential on the helix surface (Fig 2A)—and its hydrophobic residues contributing to the overall stability of the protomer (Fig 7D). Therefore, the interface that is used for trimerization is essentially composed of acidic residues. Remarkably, residues D263, E267, E274, E281, D285, and E288 are aligned on the same side of the helix (Fig 7D). We reconstructed MOKV G post-fusion trimer by modeling the arrangement of the helices at the trimeric interface. For this purpose, we superimposed them on a GCNt coiled coil [35,36] (Fig 7E). In this model of the post-fusion trimer, there are no steric clashes between subunits (which was not the case when we used VSV G post-fusion trimer as a model) (S5 Fig). We observed that the acidic residues that are aligned on the helix indeed point toward the trimer axis (Fig 7E). Therefore, they have to be protonated to stabilize the post-fusion trimer through hydrogen bonds.

As the post-fusion trimer still makes up the majority of the population up to pH 6.4 [10,13], the pKa of the lateral chain of those acidic residues (which generally is ~ 3.8 for aspartic residues and ~ 4.5 for glutamic residues that are solvent-exposed) has to be shifted toward higher values in the post-fusion trimer environment. Indeed, computation of pKa values of MOKV G aspartate and glutamate using the PROPKA 2.0 server reveals (i) a pKa value of E269 (facing D211 in the post-fusion monomer) which is above nine and (ii) a large increase of the pKa values for residues D275, D285, E267, and E274 in the modeled trimer compared to the monomeric form (S6 Fig).

## Discussion

The X-ray structure of MOKV $G_{1-436}^{*}$ in its post-fusion state described here provides insights into the architecture of the lyssaviruses glycoproteins. The comparison with VSV G and CHAV G reveals that the global fold of G is well conserved across the rhabdovirus family. The PHD, which contains antigenic site II, is by far the most divergent domain. In the TrD, the helix and its adjacent β-sheet, which constitute the internal part of the trimeric structures, are much more conserved than the lateral part, which is solvent-exposed and contains antigenic site III. Globally, the structure reveals that the major antigenic sites are located at the top of the molecule in its pre-fusion conformation and that, at the surface of the virion, they are very accessible to antibodies. Their low conservation (compared to the rest of G sequence) also indicates that the humoral response is at the origin of the diversification of rhabdovirus glycoprotein architecture. In keeping with this idea, the FD is very conserved and does not contain antigenic sites. In fact, the spikes organization at the surface of rhabdoviruses at high pH, and particularly the dense layer they form for lyssaviruses, protects the FD from being a target for the humoral response.

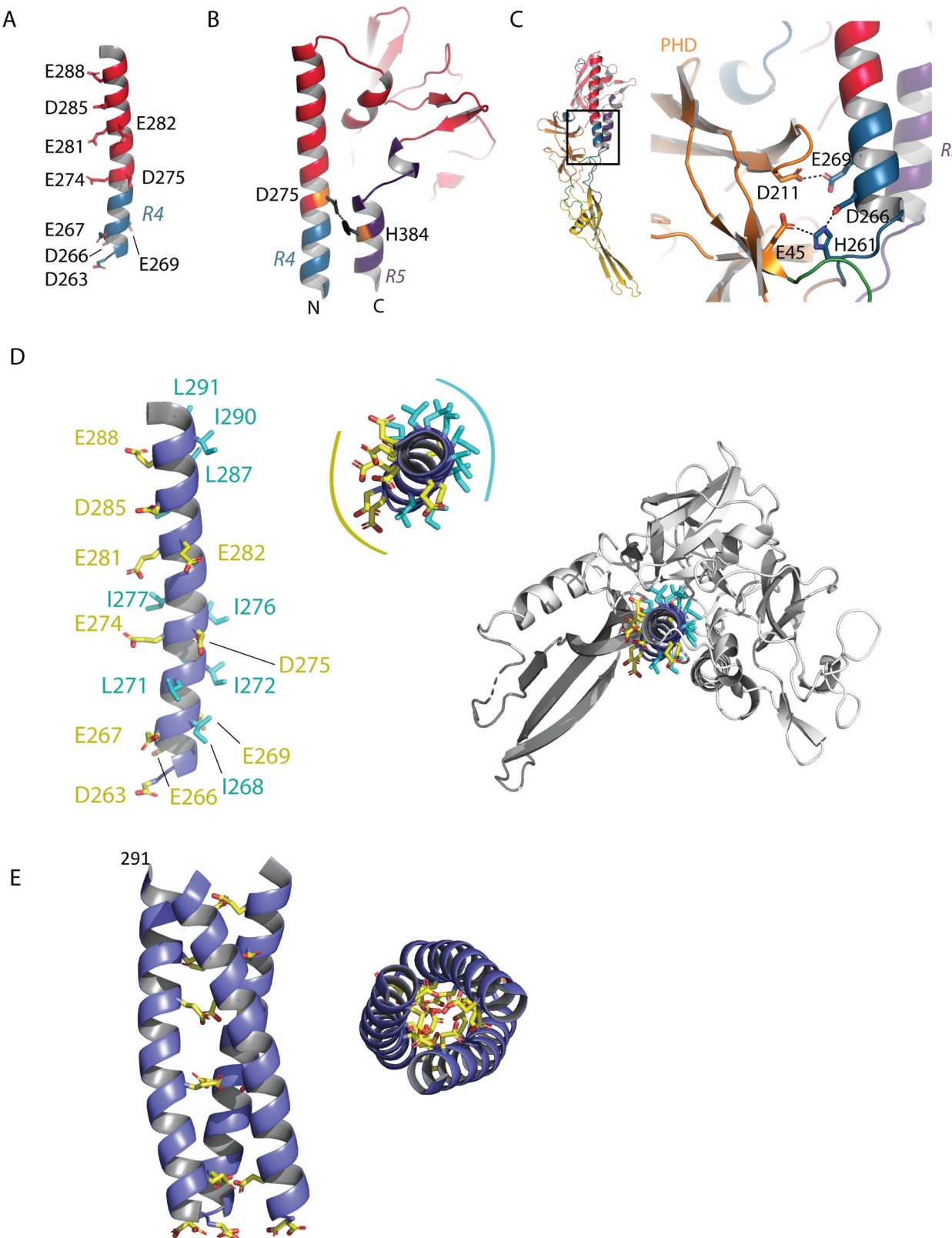

**Fig 7. pH-sensitive molecular switches in MOKV G.** A) Position of acidic residues on the TrD long helix. B) Lateral view of the TrD and segments R4 and R5 showing the interaction between D275 and H384 stabilizing the post-fusion state of MOKV G. C) Close up view on the pair of acidic residues (D211 in PHD and E269 in R4) facing each other in the post-fusion conformation of MOKV G and on the environment of H261 (in R4), which is sandwiched by D266 (in R4) and E45 (in PHD). D) View of the TrD long helix showing the disposition of hydrophobic (in cyan) and acidic

(in yellow) residues. Residues D263, E267, E274, E281, D285, and E288 are aligned on the same side of the helix of which the top view reveals the amphiphilic character. On the right, the top view indicates that the hydrophobic residues are buried in the structure and contribute to its overall stability. E) Model of the arrangement of the helices at the trimeric interface. The helices were superimposed on a GCNt coiled-coil [35, 36]. In this model, the acidic residues that are aligned on the helices point toward the trimer axis.

The presence of lysine and arginine in positions 330 and 333 respectively of RABV G is required for both RABV G interaction with the low-affinity nerve growth factor receptor (P75NTR) [6] and RABV penetration into the neurons of adult mice [22]. In the model of the pre-fusion state based on VSV G, residues K330 and D333 of MOKV G are in the vicinity of the VSV G interaction site with CR domains from members of the LDL-R family [29] (Fig 6D), which suggests that the same region of G is used by rhabdoviruses to bind their receptor. As this region is part of antigenic site III, this suggests that the humoral immune response also contributes to receptor diversification among the rhabdovirus family. In support of this notion, it has been shown that MOKV G does not recognize P75NTR [37]. It is not known whether MOKV G recognizes the other receptors (i.e., the nicotinic acetylcholine receptor) [4,38], the neural cell adhesion molecule [5], and the metabotropic glutamate receptor subtype 2 [7]) that have been proposed for RABV. For those receptors, the binding site on RABV G is not identified.

Although MOKV G has been crystallized in a monomeric conformation, EM reveals that, at low pH, it adopts a trimeric post-fusion conformation at the viral surface, which indicates once again that for both lyssaviruses and vesiculoviruses the trimer is less stable than class I fusion glycoproteins, as largely documented in the literature [11,39]. This is in agreement with the model that, for rhabdoviruses, the structural transition proceeds through monomeric intermediates [27,40].

In the case of VSV G, on the central helix, there is a single acidic (D268) residue pointing toward the trimer axis [24,34]. For MOKV G, there are six acidic residues, placed all along the central helix at the positions classically occupied by leucine and isoleucine in canonical heptad repeats (Fig 7D). Those residues are largely conserved among lyssaviruses (S2 Fig). They must be protonated in the post-fusion trimer as their deprotonation immediately induces repulsive forces that destabilize the trimeric interface. Computation of their pKa indeed reveals a shift toward higher values in the post-fusion trimer environment. These higher pKa values explain why the post-fusion trimer is still stable at pH 6.4. This also explains the strong cooperativity of the structural transition when the pH is lowered [13], as the post-fusion state has a stronger affinity for H$^+$ ions.

EM experiments reveal the ability of MOKV G to form a network at the virion surface when it adopts its post-fusion trimeric conformation. Such a behavior has been also observed for VSV G [28,34]. The lateral interactions leading to the network formation probably involve contacts between the top of the molecules in their post-fusion conformation. They do not require the presence of the transmembrane or intraviral domain as, at low pH, MOKV G ectodomain is independently able to self-associate into a quasi-crystalline array at the surface of liposomes forming elongated proteolipidic tubes very similar to those formed by VSV G ectodomain [28], which, once again, reveals the ability of rhabdovirus glycoproteins to deform membranes. This low pH-induced local reorganization of the glycoprotein network at the surface of the virion, outside the contact zone with the target membrane, might drive pore enlargement as suggested for VSV G [28]. The formation of such networks of spikes in their post-fusion conformation is very general among class III fusion glycoproteins as it has been also observed with pseudorabies virus fusion glycoprotein gB [41]. It is also reminiscent of the behavior of class II fusion glycoproteins, which, in their post-fusion conformation, form more or less regular networks that are different from their initial icosahedral organization [42–44].

In conclusion, this work describes the first structure of a lyssavirus glycoprotein, which paves the way for rational mutagenesis. By delimiting structural domains and antigenic sites, it should help to design chimeric lyssavirus glycoproteins, which may be useful for the development of a pan-lyssavirus vaccine.

## Material and methods

### Chemicals

Phosphatidylcholine (PC; type XVI-E from egg yolk), phosphatidylethanolamine (type IV from soybean), cholesterol (Chol; standard for chromatography), and sphingomyelin (SM; from chicken egg yolk) were supplied by Sigma-Aldrich. Tb-Xo4 was purchased from Polyvalan Lyon, France [45].

### Cells and viruses

HEK-293T (human embryonic kidney cells expressing simian virus 40T antigen; ATCC CRL-3216) cells were grown in Dulbecco's modified Eagle's medium (DMEM) supplemented with 10% fetal calf serum (FCS). Cells were maintained at 37˚C in a humidified incubator with 5% $CO_2$. *Drosophila* Schneider 2 (S2) cell line (Invitrogen) were grown in synthetic Express Five SFM media (Gibco) supplemented with 20 mM L-Glutamine and 50 U/mL of penicillin and 50 μg/mL of streptomycin. Cells were maintained at 28˚C.

VSVΔG-eGFP is a recombinant VSV which was derived from a full-length cDNA clone of the VSV genome (Indiana serotype) in which the coding region of the G protein was replaced by a modified version of the GFP gene and pseudotyped with the VSV G protein [34,46,47]. VSVΔG-eGFP is propagated on HEK-293T cells that have been previously transfected with pCAGGS-VSV G.

### Plasmids and cloning

Genes encoding MOKV G WT (GenBank: Y09762.1) and MOKV G* (in which the hydrophobic fusion loops [74]FVGY[77] and [121]WL[122] have been replaced by more hydrophilic sequences [74]GNGS[77] and [121]GS[122]) were ordered from Eurofin Genomics. For transient expression in HEK-293T cells, MOKV G WT and MOKV G* genes were subcloned into pCAGGS vector using Gibson assembly reaction kit (New England Biolabs). For stable expression in S2 cells, the ectodomain sequences MOKV $G_{1-436}$WT and MOKV $G_{1-436}$* were subcloned into the expression vector pT350 [48] using Gibson assembly reaction kit (New England Biolabs) so that MOKV G ecdomains were flanked by the *Drosophila* Bip secretion signal at the N-terminus and an enterokinase-cleavable double StrepTag at the C-terminus (sequence: DDDDKAGWSHPQFEKGGGSGGGSGGGSWSHPQFEK).

### MOKV $G_{1-436}$ production and purification

As explained in Backovic and Krey [48], stable cell lines expressing the MOKV $G_{1-436}$WT or MOKV $G_{1-436}$* proteins with a C-terminal tandem StrepTag were generated by cotransfecting S2 cells with a plasmid encoding MOKV $G_{1-436}$WT or MOKV $G_{1-436}$* in a pT350 vector and a plasmid encoding the puromycin resistance gene. To produce MOKV $G_{1-436}$WT or MOKV $G_{1-436}$* in large quantities, the S2 cells were grown in suspension at 28˚C and 120 rpm until reaching $1.5 \ 10^7$ cells/ml in media supplemented with 7μg/ml puromycin. The cell culture was then diluted twice and induced with 500μM of $CuSO_4$. The culture was harvested five days after induction, and the supernatant was concentrated through tangential filtration (Vivaflow 200, VWR). MOKV $G_{1-436}$WT or MOKV $G_{1-436}$* were purified thanks to their StrepTag on a

streptavidin affinity column (StrepTrap, GE Healthcare) in a 20 mM Tris-HCl pH 8 buffer, 150 mM NaCl, and 2 mM EDTA. Elution was carried out in the same buffer supplemented with 3 mM α-Desthiobiotin. For MOKV $G_{1-436}$*, an additional purification step was performed on a size exclusion chromatography (Superdex 200 increase column 10/300, GE Healthcare) equilibrated in 20mM Tris-HCl pH 8, 150mM NaCl, and 2mM EDTA. Purified MOKV $G_{1-436}$WT or MOKV $G_{1-436}$* were concentrated (Amicon Ultra 30kDa c / o -Millipore-) and stored at -80˚C.

## Preparation of pseudotypes

HEK-293T cells at 80% confluence were transfected by pCAGGS encoding MOKV G WT or MOKV G WT * using PEI Max (Polyscience). Twenty-four hours after transfection, cells were infected with VSVΔG-eGFP and pseudotyped with VSV G at an MOI of 1. At two hours p.i., cells were washed to remove residual viruses from the inoculum. Cell supernatants containing the pseudotyped viral particles were collected at 16 h.p.i. Viral particles were purified and concentrated as previously described [28].

## Preparation of liposomes

We mixed 250 μg of PC, 250 μg of PE, 250 μg of SM, and 375 μg of Chol dissolved in organic solvents and dried the mixture under vacuum. The lipid film was resuspended in 1 ml of buffer (5 mM Tris-HCl pH 8, 150 mM NaCl). The mixture was subjected to three cycles of freezing/thawing in liquid nitrogen and then sonicated in a water bath for 20 min.

## Negative staining electron microscopy

Purified pseudotyped virions were diluted in 20 mM HEPES pH 7.5 and 150 mM NaCl. The pH was progressively lowered by dialyzing virions overnight at 37˚ C against the same buffer in which 100 mM Ammonium Acetate pH 5.5, 375 mM NaCl was gradually added. Samples at pH 7.5 or 5.5 were adsorbed onto airglow discharge carbon-coated grids and stained with sodium phospho-tungstic acid adjusted to the sample pH.

Purified MOKV $G_{1-436}$* and VSV $G_{th}$ (purified as in [28]) were diluted in 150 mM NaCl in either 50 mM Tris-HCl pH 7.5, or 50 mM MES at pH 5.5. Proteins were then adsorbed onto grids and stained with sodium phospho-tungstic acid adjusted to the sample pH.

Purified MOKV $G_{1-436}$WT was incubated with liposomes at 37˚C in 150 mM NaCl, 50 mM MES at pH 5.5. The mixture was then adsorbed onto grids and stained with sodium phospho-tungstic acid adjusted to the sample pH or uranyl acetate (pH 4.5).

Images were recorded in an electron microscope Tecnai Spirit (FEI) operated at 100kV.

## Electron cryomicroscopy

We dispensed 3 μl of the studied suspension (MOKV $G_{1-436}$WT was incubated with liposomes at 37˚C in 150 mM NaCl, 50 mM MES at pH 5) on a holey carbon grid (Quantifoil R2/2). The drop was blotted, and the grid was plunged into liquid nitrogen-cooled ethane and transferred to a liquid nitrogen-cooled holder (model 626; Gatan). Samples were observed in an electron microscope (TECNAI FEG 200kV FEI) operating at 200 kV equipped with a K2 Summit direct detection camera. Images were recorded under minimal dose conditions at a magnification of 19,000.

## Crystallization and data collection

Commercial kits (The Classics, The MbClass, The MbClass I and II, The PEGs I and II screens from Qiagen/Nextal) containing 96 different precipitants were used to screen the crystallization conditions of MOKV $G_{1-436}^*$. Initial crystallization experiments were performed at 20˚ C by sitting drop vapor diffusion (100 nl protein and 100 nl mother liquor) in 96-well TTP plates (Corning) using a Mosquito robot. Crystals of MOKV $G_{1-436}^*$ were grown in hanging drops containing equal volumes (1 μL) of purified proteins and reservoir solution (12% PEG 4000, 100 mM HEPES pH 7.5, 100 mM Sodium acetate) in the presence of 10 mM Tb-Xo4 crystallophore. Crystals were observed after 3–5 days and matured to full size within two weeks. They were then soaked into mother liquor supplemented with 35% (v/v) glycerol for cryo-protection. Subsequently, crystals were plunged into liquid nitrogen and measured at the SOLEIL synchrotron beam line PROXIMA-1. A single data set was collected at 100 K at the terbium LIII absorption edge on a single crystal up to 2.9 Å resolution.

## Structure determination and refinement

Diffraction data were integrated and reduced using XDS program package [49]. The crystals of MOKV G1-436* belonged to the C2221 space group. As crystals of MOKV G were anisotropic, data were submitted to the STARANISO server (http://staraniso.globalphasing.org/cgi-bin/staraniso.cgi) to apply an anisotropy correction. These corrected data were then used for the refinement. We used the terbium anomalous signal of Tb-Xo4(III) used in co-crystallization to solve the structure by the SAD method. A unique molecule was found in the asymmetric unit with a solvent content of 62%, and six Tb-Xo4 were found. Initial phases and models were obtained using the SHELX C/D/E suite of programs [50]. Subsequently, phases were calculated using PHASER [51] and improved by solvent flattening using PARROT [52]. Improved models were automatically built using Buccaneer [52] and then iteratively rebuilt using COOT [53] and refined using Phenix [54]. The details of the crystallographic analysis are presented in Table 1. Structure representations were made using PyMol (PyMOL Molecular Graphics System. DeLano Scientific LLC, San Carlos, CA, USA. http://www.pymol.org).

## Model determination of MOKV G pre-fusion protomer and MOKV G post-fusion trimer

The model of MOKV G pre-fusion protomer was obtained by structural superposition of MOKV G FD, TrD, and PHD on VSV G pre-fusion protomer (PDB code: 5I2S) [24] using the align command in PyMOL.

The model of MOKV G post-fusion trimer was obtained by positioning three copies of MOKV G helix 264–294 on the coiled-coil motif protein of parainfluenza virus 5 F protein fused to a GCNt trimeric coiled-coil domain (PDB code: 2B9B) [36].

## Supporting information

**S1 Fig. SDS PAGE analysis of purified MOKV $G_{1-436}$ and $G_{1-436}^*$ after the streptavidin affinity column step.**
(TIF)

**S2 Fig. Sequence alignment of MOKV G and VSV G ectodomain based on the crystal structures.** Conserved residues are in red boxes, whereas similar ones are in pink. Domains are identified by colors following Table 2 color code. Secondary structures of MOKV G and VSV G post-fusion state are indicated by arrows for β-sheets and straight lines for helices.
(TIF)

**S3 Fig. Sequence alignment of lyssavirus glycoproteins.** Conserved residues are in red boxes, whereas similar ones are in pink. A square indicates residues that are found substituted in mutants escaping neutralization by MAbs (in magenta for antigenic site II, in brown for antigenic site III, in cyan for minor site a, and in green for low pH antigenic site). The location of the 17D2 antigenic site is underlined in pink. Black stars indicate acidic residues in the long helix. Purple stars indicate the other acidic residues or the histidines that play the role of pH-sensitive conformational switches (see text).
(TIF)

**S4 Fig. Sequence alignment of MOKV G and RABV G ectodomain.** Conserved residues are in red boxes, whereas similar ones are in pink. Colors following Table 2 color code identify domains. Secondary structures of the crystalline MOKV G post-fusion monomer are indicated by arrows for β-sheets and straight lines for helices.
(TIF)

**S5 Fig. Lateral and top views of the modeled MOKV G post-fusion trimer.** A) MOKV G post-fusion trimer was constructed by modeling the arrangement of the helices at the trimeric interface. For this purpose, the helices were superimposed on a GCNt coiled-coil. In this post-fusion trimer model, there are no steric clashes between subunits. The fusion domains slightly split apart, which may not be the case in the real post-fusion trimer (due to flexibility of R2 and R3 segments or of the fusion domain itself). B) MOKVG post-fusion trimer was constructed by superimposing the Trd domains of MOKV G on the VSV G post-fusion trimer. Steric clashes are visible in top view (showing promiscuity of the three central helices) and in side view (showing important clashes between fusion domains).
(TIF)

**S6 Fig.** computed pKas of aspartic (A) and glutamic (B) residues in the context of MOKV G monomer plotted against their computed pKa in the modeled post-fusion trimer. Residues exhibiting an important shift (compared to the pKa of Asp and Glu exposed to solvent are numbered). The pKa increase for residue Glu 212 is due to the fact that, in the modeled trimer, it faces Asp 386 of the neighboring protomer.
(TIF)

## Acknowledgments

We acknowledge synchrotron SOLEIL (Saint-Aubin, France) for providing radiation facilities and the Structural biology pole of the IMAGIF integrated platform (https://www.imagif.cnrs.fr/?nlang=en) for access to crystallization and electron microscopy facilities.

## Author Contributions

**Conceptualization:** Aurélie A. Albertini.

**Data curation:** Stéphane Roche, Pierre Legrand.

**Formal analysis:** Laura Belot, Stéphane Roche, Yves Gaudin, Aurélie A. Albertini.

**Funding acquisition:** Yves Gaudin.

**Investigation:** Laura Belot, Malika Ouldali, Aurélie A. Albertini.

**Methodology:** Pierre Legrand.

**Supervision:** Yves Gaudin, Aurélie A. Albertini.

**Validation:** Yves Gaudin, Aurélie A. Albertini.

**Writing – original draft:** Yves Gaudin, Aurélie A. Albertini.

**Writing – review & editing:** Laura Belot, Stéphane Roche, Yves Gaudin, Aurélie A. Albertini.

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
