## [Decision Letter · Decision Letter 0]

13 Jan 2020

Dear Mr Gaudin:

Thank you very much for submitting your manuscript "Crystal structure of Mokola Virus Glycoprotein in its post-fusion conformation" (PPATHOGENS-D-19-02259) for review by PLOS Pathogens. Your manuscript was fully evaluated at the editorial level and by independent peer reviewers. 

As you can see from the attached reviews, all three reviewers appreciated your study and its importance. There were several issues, which can be quickly addressed.  Concerns were some “overstatements” of the impact of the research and clarification. These overstatements should be toned down. One reviewer also recommended some careful editing to remove and/or improve several instances of non-standard English usage; this is important for the greater clarity of your work.  Other reviewers made helpful suggestions for some further explanation(s), and small additions, which the authors should judiciously follow. However, I do agree with the reviewer that this is a critical study, and I am delighted that you consider PLoS Pathogens for your manuscript.

We therefore ask you to modify the manuscript according to the review recommendations before we can consider your manuscript for acceptance. Your revisions should address the specific points made by each reviewer.

(1) A letter containing a detailed list of your responses to the review comments and a description of the changes you have made in the manuscript. Please note while forming your response, if your article is accepted, you may have the opportunity to make the peer review history publicly available. The record will include editor decision letters (with reviews) and your responses to reviewer comments. If eligible, we will contact you to opt in or out.

(2) Two versions of the manuscript: one with either highlights or tracked changes denoting where the text has been changed; the other a clean version (uploaded as the manuscript file).

We hope to receive your revised manuscript within 60 days or less. If you anticipate any delay in its return, we ask that you let us know the expected resubmission date by replying to this email.

[LINK]

Sincerely,

Matthias Johannes Schnell, PhD

Associate Editor

PLOS Pathogens

Benhur Lee

Section Editor

PLOS Pathogens

Kasturi Haldar

Editor-in-Chief

PLOS Pathogens

orcid.org/0000-0001-5065-158X

Michael Malim

Editor-in-Chief

PLOS Pathogens

orcid.org/0000-0002-7699-2064

Reviewer's Responses to Questions

**Part I - Summary**

Reviewer #1: Although rabies virus and emerging additional pathogens of the lyssavirus genus, native structural information for the viral glycoprotein is currently available only for members of the related visiculovirus genus (VSV and Chandipura virus) but not for any lyssavirus. This manuscript by Belot and colleagues makes an important first contribution to closing this knowledge gap, reporting the structure of a modified Mokola virus G protein monomer in postfusion conformation. Data are overall of good quality and appropriately presented. A strength of the work is the inclusion of a recombinant VSV-delta G pseudotyped with either wild type or equally fusion loop-modified MOKV G, supporting that the fusion loop modification does not interfere with G trimerization.

This work advances the field, but does not benefit from, or live up to, the repeated overreaching translational claims of “[paving] the way for the development of a pan-lyssavirus vaccine” or use of RABV G “for transneuronal tracing and studies of monosynaptic connections”. These unsubstantiated statements distract from appreciation of the true impact of the work and need to be removed, or appropriately clarified.

Reviewer #2: This paper describes the first structure of a lyssavirus glycoprotein, which is important for the development of a pan-lyssavirus vaccine. Overall, the paper reads well and provides a significant contribution to the field.

I invite the authors to address my comments detailed below:

Reviewer #3: Crystal structure of Mokola virus glycoprotein in its post-fusion conformation. PPATHOGENS-D-19-02259

This paper by Laura Belot et al presents the first lyssavirus glycoprotein structure in its post-fusion conformation. The work is very well conducted, the experiments nicely designed and undertaken. The I2BC laboratory is and expert team on solving Rhabdoviruses G protein structures and they use their art skills to underpin interesting features about the MOKV G protein structure presented: localization of the antigenic sites, post-fusion trimers, pH sensitive molecular switches.

**Part II – Major Issues: Key Experiments Required for Acceptance**

Reviewer #1: 1) 45-46; 61-62; 340: These statements need to be toned down, or substantiated. This structure provides an appreciation of antigenic sites (nicely discussed), but even three-times repetition does not change the fact that “paving the path” is too strong. Antigenic site mapping is based on repositioning of individual MOKV G structural domains on prefusion VSV G, which appears to be a structure guided form of homology modeling (see comment #5), but homology modeling nevertheless. The only actual discussion of the pan-vaccine idea reaches the conclusion that the FD is highly conserved, but makes a poor antibody target (296-299). Although this is certainly correct, this knowledge predates this study and the MOKV G structure adds very little, if anything, to solve the pan-lyssavirus vaccine problem.

2) 136: Please back up the statement re purity of the protein preparations, i.e. by showing total protein stains after SDS-PAGE.

3) Can you support your statements in the paragraph starting at line 200 (Fig 5) with 2D class-averaging?

4) 319-320: The GCN4-based postfusion trimer model predicts protonation of at least six acidic residues in the central helix, requiring a pH drop to 5.5-6.0 to be sufficient to result in mass-protonation of aspartate and glutamate side chains at positions 263, 267, 274, 281, 285, and 288. Please provide some detail on the basis and nature of the proposed pKa shift, and add simulations to predict the protonation states of these residues at the appropriate pH points (5.5 and 7.5).

5) 310: There is insufficient data support for this claim. Take multiple measurements, state the number measurements performed, and add class averaging.

6) The tubular structures shown in figure 4B are not very informative as presented. Can you perform helical reconstructions to determine the regular parameters of these helical arrays?

Reviewer #2: (No Response)

Reviewer #3: Nevertheless some questions or points are still raised:

1) The sentence “This work paves the way for the development of a pan-lyssavirus vaccine. Furthermore, as RABV G is used for transneuronal tracing and studies of monosynaptic connections, the knowledge of MOKV G structure might be useful to improve the available tools in this field” is highly speculative at this stage since no experimental data are provided to sustain. Please remove this from the author summary.

2) Identically, the sentence “Globally, this first lyssavirus G structure paves the way for the development of a pan-lyssavirus vaccine” is a hope at this stage.

Both sentences 1) and 2) could be discussed further in the discussion.

3) The authors used two G constructs G(1-436)WT and G(1-436)**. They state that G(1-436)WT was less expressed and its purification was less optimal. Anyhow, they use it for the liposomes experiments. Please provide all the data for the expression and the purification for both constructs.

4) Fig 1C is difficult to read. Can the authors increase the quality and/or the magnification?

5) Line 150: “We screened several crystallization conditions….” Please provide more details in the experimental procedures.

6) Paragraph line 180: the authors use G(1-436)WT for the liposomes experiments but they did not try to crystallize it. Could they comment their choice? Having solved the structure of G(1-436)**, don’t they think that some information on G(1-436) could be useful to strengthen their work?

7) To investigate further the biological activity of MOKV G protein, the authors used the VSVΔG-eGFP virus. It is certain interesting to show the pseudotyping feasibility between vesiculovirus and lyssavirus but in this case RABVΔG would have been much more informative. Especially since they observed that the pre-fusion state of G(1-436)** is partially destabilized. I would suggest that experiments with RABVΔG are provided.

8) S2 Fig, even though comparison of all the lyssavirus is interesting, most of the Mab work has been undertaken with RABV. Please provide also a fig with MOKV and RABV only for clarity.

9) Line 300: only P75NTR is discussed. Please add the other receptors in the discussion as far as it is possible.

10) Paragraph line 339: According to point 1) authors should discuss further how their structure can help on these matters.

**Part III – Minor Issues: Editorial and Data Presentation Modifications**

Reviewer #1: 7) 93: “Wavy” does not come across as the best choice of descriptor. I wonder whether you can rephrase the description of the virion surface.

8) 222: How was this “repositioning” performed specifically? This strategy appears to be absent from Methods, and without a detailed understanding of the approach this section is not fully convincing.

9) Please discuss how and why the remarkable diversity of the TrD domain may have evolved. A whole panel (figure 3C) is dedicated to this observation, but discussion of the point remains scarce.

10) 609: Please add a postfusion trimer model based on VSV G and substantiate the claimed “sterical clashes” in a side-by-side comparison in figure S3 to illustrate the point for the reader.

11) The manuscript needs language editing throughout.

Reviewer #2: Abstract and Results (Line 281): “At high pH, when they are negatively charged, they destabilize the interface, which explains the conformational change reversibility”.

What does mean ‘at high pH’? the pKa of Asp and Glu is ~4.75. These residues will be negatively charged at pH > 4.7. Clarify what ‘high’ pH stands for here.

Line 109: “Sequence alignments indicate that lyssaviruses glycoproteins adopt the same fold as

other rhabdovirus glycoproteins and therefore belong to the third class (class III) of viral fusion glycoproteins”.

Rephrase. Sequence alignment cannot ‘indicate’ the fold of a protein. Refer instead to sequence similarity and identity.

Line 134: I find the nomenclature used in this paper (MOKV G1-436**) with two asterisks to be awkward. Perhaps the two asterisks can be replaced with something else.

Line 311: “lyssaviruses and vesiculoviruses, the trimer is unstable”.

Can the authors describe why the trimer is unstable?

Figure 1C. Right panel. VSV Gth at pH 5.5. Can the authors rule out this protein is not just an aggregate commonly seen by negative stain?

Figure 2. It is difficult to decipher what is new (and presented for the first time in this paper) from previously solved structures. Please, dedicate Figure 2 to describe the structure of MOKV G1-436**, then use another figure, or subpanels of Fig 2 to compare MOKV G1-436** to known fusion proteins. Refer to the sequence similarity/identity and RMSD after secondary structure superimposition of MOKV G1-436** with other viral glycoproteins. Also, describe the surface potential, charge, etc. of MOKV G1-436**.

Figure 7. The paper lacks a 3D-model for the trimeric lyssavirus glycoprotein. Can this be generated using the structural information provided in the work?

Crystallographic table: The I/SI in the outer shell is very low: 0.95, which means the error exceeds the measurement. I find it hard to believe the CC1/2 in the outer shell is 0.798. Also, please indicate the CC1/2 as 0.998 (0.798). Finally, include an Rsym.

Reviewer #3: None

PLOS authors have the option to publish the peer review history of their article (what does this mean?). If published, this will include your full peer review and any attached files.

Reviewer #1: No

Reviewer #2: No

Reviewer #3: No

---

## [Editor Report · Decision Letter 1]

5 Feb 2020

Dear Mr Gaudin,

We are pleased to inform you that your manuscript 'Crystal structure of Mokola Virus Glycoprotein in its post-fusion conformation' has been provisionally accepted for publication in PLOS Pathogens.

Before your manuscript can be formally accepted you will need to complete some formatting changes, which you will receive in a follow up email. A member of our team will be in touch within two working days with a set of requests.

Best regards,

Matthias Johannes Schnell, PhD

Associate Editor

PLOS Pathogens

Benhur Lee

Section Editor

PLOS Pathogens

Kasturi Haldar

Editor-in-Chief

PLOS Pathogens

orcid.org/0000-0001-5065-158X

Michael Malim

Editor-in-Chief

PLOS Pathogens

orcid.org/0000-0002-7699-2064
---

## [Editor Report · Acceptance letter]

4 Mar 2020

Dear Mr Gaudin,

We are delighted to inform you that your manuscript, "Crystal structure of Mokola Virus Glycoprotein in its post-fusion conformation," has been formally accepted for publication in PLOS Pathogens.

Best regards,

Kasturi Haldar

Editor-in-Chief

PLOS Pathogens

orcid.org/0000-0001-5065-158X

Michael Malim

Editor-in-Chief

PLOS Pathogens

orcid.org/0000-0002-7699-2064